# A Bridging Framework for Model Optimization and Deep Propagation

**Risheng Liu**[1,2]*, **Shichao Cheng**[3], **Xiaokun Liu**[1], **Long Ma**[1], **Xin Fan**[1,2], **Zhongxuan Luo**[2,3]

[1]International School of Information Science & Engineering, Dalian University of Technology
[2]Key Laboratory for Ubiquitous Network and Service Software of Liaoning Province
[3]School of Mathematical Science, Dalian University of Technology

## Abstract

Optimizing task-related mathematical model is one of the most fundamental methodologies in statistic and learning areas. However, generally designed schematic iterations may hard to investigate complex data distributions in real-world applications. Recently, training deep propagations (i.e., networks) has gained promising performance in some particular tasks. Unfortunately, existing networks are often built in heuristic manners, thus lack of principled interpretations and solid theoretical supports. In this work, we provide a new paradigm, named Propagation and Optimization based Deep Model (PODM), to bridge the gaps between these different mechanisms (i.e., model optimization and deep propagation). On the one hand, we utilize PODM as a deeply trained solver for model optimization. Different from these existing network based iterations, which often lack theoretical investigations, we provide strict convergence analysis for PODM in the challenging nonconvex and nonsmooth scenarios. On the other hand, by relaxing the model constraints and performing end-to-end training, we also develop a PODM based strategy to integrate domain knowledge (formulated as models) and real data distributions (learned by networks), resulting in a generic ensemble framework for challenging real-world applications. Extensive experiments verify our theoretical results and demonstrate the superiority of PODM against these state-of-the-art approaches.

## 1   Introduction

In the last several decades, many machine learning and computer vision tasks have been formulated as the problems of solving mathematically designed optimization models. Indeed, these models are the workhorse of learning, vision and power in most practical algorithms. However, it is actually hard to obtain a theoretically efficient formulation to handle these complex data distributions in different practical problems. Moreover, generally designed optimization models  [3, 5] may be lack of flexibility and robustness leading to severe corruptions and errors, which are commonly existed in real-world scenarios.

In recent years, a variety of deep neural networks (DNNs) have been established and trained in end-to-end manner for different learning and vision problems. For example, AlexNet [13] first demonstrated the advantages of DNNs in the challenge of ImageNet large scale visual recognition. With a careful design,  [29] proposed GoogleNet, which increased the depth and width of the network while keeping the computational budget constant. However, some researchers also found that although increasing the layers of the networks may improve the performance, it is more difficult to train a deeper network. By introducing shortcut blocks,  [10] proposed the well-known residual network. It has been verified that the residual structure can successfully avoid gradient vanishing problems and thus significantly

improve the practical training performance for deeper network. Besides the great success of DNNs in supervised learning, some efforts have also been made on unsupervised learning tasks. [8] proposed the generative adversarial network, which utilizes a pair of generator network and discriminator network contesting with each other in a zero-sum game framework to generate the realistic samples. Though with relatively good performance on specific applications, the interpretability issue is still a big problem for existing DNNs. That is, it is challenging to reason about what a DNN model actually does due to its opaque or black-box nature.

Embedding DNNs into the optimization process is recently popular and some preliminary works have been developed from different perspectives. For example, [9] trained a feed-forward architecture to speed up sparse coding problems. [1] introduced deep transformations to address correlation analysis on multiple view data. Very recently, to better address the true image degradation, [7, 33, 30] incorporated convolutional DNN as image priors into the maximum a posterior inference process for image restoration. Another group of recent works also tried to utilize recurrent neural network (RNN) structures [2] and/or reinforcement strategies [17] to directly learn descent iterations for different learning tasks. It should be pointed out that the convergence issue should be the core for optimization algorithm design. Unfortunately, even with relatively good practical performance on some applications, till now it is still challenging to provide strict convergence analysis on these deeply trained iterations.

## 1.1 Our Contributions

As discussed above, the interpretability and guarantees are the most important missing footstones for the previous experience based networks. Some preliminary investigations have been proposed to combine numerical iterations and learnable architectures for deep propagations design. However, due to these naive combination strategies (e.g., directly replace iterations by architectures), it is still challenging to provide strict convergence analysis on their resulted deep models. To partially break through these limitations, this paper proposes a theoretically guaranteed paradigm, named Propagation and Optimization based Deep Model (PODM), to incorporate knowledge-driven schematic iterations and data-dependent network architectures to address both model optimization and learning tasks. On the one hand, PODM actually provides a learnable (i.e., data-dependent) numerical solver (See Fig. 1). Compared with these naive unrolling based methods (e.g., [7, 33, 30, 17, 2, 20]), the main advantage of PODM is that we can generate iterations, which strictly converge to the critical point of the given optimization model, even in the complex nonconvex and nonsmooth scenarios. On the other hand, by slightly relaxing the exact optimality constraints during propagations, we can also obtain an *interpretable* framework to integrate mathematical principles (i.e., formulated by model based building-block) and experience of the tasks (i.e., network structures designed in heuristic manners) for *collaborative* end-to-end learning.

In summary, the contributions of this paper mainly include:

- We provided a model-inspired paradigm to establish building-block modules for deep model design. Different from existing trainable iteration methods, in which the architectures are built either from specific prior formulations (e.g., Markov random fields [24]) or completely in heuristic manners (e.g., replace original priors by experience based networks [7, 33]), we develop a flexible framework to integrate both data (investigated from training set) and knowledge (incorporated into principled priors) for deep propagations construction.

- By introducing an optimality error checking condition together with a proximal feedback mechanism, we prove in theory that the propagation generated by PODM is globally[2] convergent to the critical point of the given optimization model. Such strict convergent guarantee is just the main advantage against these existing deep iterations designed in heuristic manner (e.g., [7, 33, 30, 17, 2])

- As a nontrivial byproduct, the relaxed PODM actually provides a *plug-and-play*, *collaborative*, *interpretable*, and *end-to-end* deep learning framework for real-world complex tasks. Extensive experimental results on real-world image restoration applications demonstrate the effectiveness of our PODM and its relaxed extension.

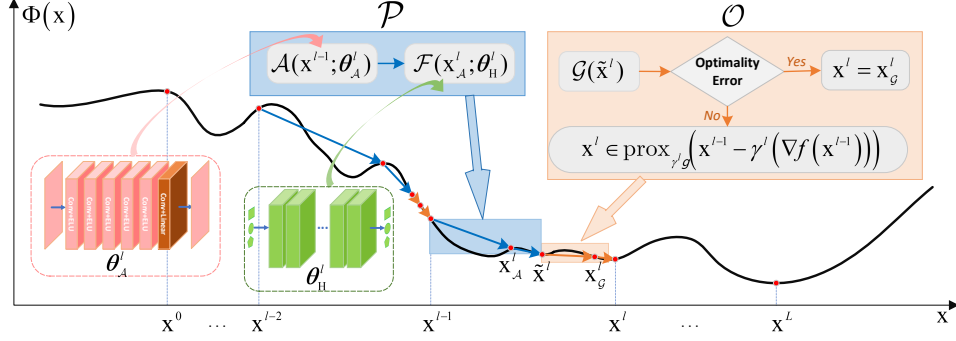

Figure 1: Illustrating the mechanism of PODM for nonconvex model optimization.

## 2 Existing Trainable Iterations: Lack Generalizations and Guarantees

We review existing training based iterative methods for model optimization. Specifically, most learning and vision tasks can be formulated as the following regularized optimization model:

$$\min_{\mathbf{x}} \Phi(\mathbf{x}) := f(\mathbf{x}) + g(\mathbf{x}), \tag{1}$$

where $f$ denotes the loss term and $g$ is related to the regularization term. Different from classical numerical solvers, which design their iterations purely based on mathematical derivations. Recent studies try to establish their optimization process based on training iterative architectures on collected training data. These existing works can be roughly divided into two categories: trainable priors and network based iterations.

The first category of methods aim to introduce hyper parameters for specific prior formulations (e.g., $\ell_1$-norm and RTF) and then unroll the resulted updating schemes to obtain trainable iterations for Eq (1). For example, the works in [9, 27] parameterize the $\ell_1$ regularizer and adopt classical first-order methods to derivate their final iterations. The main limitation of these approaches is that their schemes are established based on specific forms of priors, thus cannot be applied for general learning/vision problems. Even worse, the hyper parameters in these approaches (e.g., trade-off or combination weights) are too simple to extract complex data distributions.

On the other hand, the works in [7, 33] try to directly replace prior-related numerical computations at each iteration by experientially designed network architectures. In this way, these approaches actually completely discard the explicit regularization $g$ in their updating schemes. Very recently, the recurrent [2], unrolling [19] and reinforcement [17] learning strategies have also been introduced to train network based iterations for model optimization. Since these approaches completely discard the original regularizations (i.e., g), no prior knowledges can be enforced in their iterations. More importantly, we must emphasize that due to these embedded inexact computations, it is challenging to provide strict convergence analysis on most of above mentioned trainable iterations.

## 3 Our Model Inspired Building-Blocks

In this section, we establish two fundamental iterative modules as our trainable architectures for both model optimization and deep propagation. Specifically, the deep propagation module is designed as our generic architecture to incorporate domain knowledge into trainable propagations. While the optimization module actually enforces feedback control to guide the iterations to satisfy our optimality errors. The mechanism of PODM is briefly illustrated in Fig. 1.

Suppose $\mathcal{A}$ is the given network architecture (may built in heuristic manner) and denote its output as $\mathbf{x}_{\mathcal{A}} = \mathcal{A}(\mathbf{x}; \boldsymbol{\theta}_{\mathcal{A}})$. We would like to design our propagation module based on both $\mathcal{A}$ and $\Phi$ defined in Eq. (1). Specifically, rather than parameterizing $g$ or completely replace it by networks in existing works, we integrate these two parts by the following quadratic penalized energy:

$$\min_{\mathbf{x}} \Phi(\mathbf{x}) + d(\mathbf{x}, \mathbf{x}_{\mathcal{A}}) - \langle \mathbf{x}, \boldsymbol{\epsilon} \rangle = \min_{\mathbf{x}} \overbrace{\underbrace{f(\mathbf{x})}_{\text{Fidelity}} + \underbrace{g(\mathbf{x})}_{\text{Designed prior}}}^{\text{Knowledge}} + \overbrace{\underbrace{d(\mathbf{x}, \mathbf{x}_{\mathcal{A}})}_{\text{Learned prior}} - \underbrace{\langle \mathbf{x}, \boldsymbol{\epsilon} \rangle}_{\text{Error}}}^{\text{Data}}. \tag{2}$$

Here, $d(\mathbf{x}, \mathbf{x}_\mathcal{A})$ is the distance function which intents to introduce the output of network into the propagation module. It can be defined as $d(\mathbf{x}, \mathbf{x}_\mathcal{A}) = h(\mathbf{x}) - h(\mathbf{x}_\mathcal{A}) - \langle \nabla h(\mathbf{x}_\mathcal{A}), \mathbf{x} - \mathbf{x}_\mathcal{A} \rangle$, where $h(\mathbf{x}) = \| \cdot \|_\mathbf{H}^2$ and $\mathbf{H}$ denotes a symmetric matrix[3]. $\boldsymbol{\epsilon}$ denotes the error corresponding to Eq. (1) since introducing the network. Both the designed prior $g(\mathbf{x})$ and learned prior $d(\mathbf{x}, \mathbf{x}_\mathcal{A})$ are merged to compose our hybrid priors. Please notice that Eq. (2) can also be understood as a hybrid prior based inexact approximation of Eq. (1), in which we establish an ensemble of both domain knowledge (i.e., $g$) and training data (i.e., $\mathcal{A}$). Indeed, we can control the inexact solution by calculating the specific function about $\boldsymbol{\epsilon}$.

**Propagation Module:** We first investigate the following sub-model of Eq. (2) (i.e., only with fidelity and learned priors)

$$\mathbf{x}_\mathcal{F} = \mathcal{F}(\mathbf{x}_\mathcal{A}; \boldsymbol{\theta}_\mathbf{H}) := \arg\min_\mathbf{x} \left\{ f(\mathbf{x}) + d(\mathbf{x}, \mathbf{x}_\mathcal{A}) \right\}, \tag{3}$$

where $\boldsymbol{\theta}_\mathbf{H}$ denotes the parameter in distance $d(\mathbf{x}, \mathbf{x}_\mathcal{A})$. Eq. (3) actually integrates the principled model fidelity (i.e., $f$) and network based priors ($\mathcal{A}$). Following this formulation, we can define our data-dependent propagation module ($\mathcal{P}$) as the cascade of $\mathcal{A}$ and $\mathcal{F}$ in the $l$-th stage, i.e.,

$$\tilde{\mathbf{x}}^l = \mathcal{P}(\mathbf{x}^{l-1}; \boldsymbol{\vartheta}^l) := \mathcal{F}\left( \mathcal{A}\left(\mathbf{x}^{l-1}; \boldsymbol{\theta}_\mathcal{A}^l\right); \boldsymbol{\theta}_\mathbf{H}^l \right),$$

where $\boldsymbol{\vartheta}^l = \{\boldsymbol{\theta}_\mathcal{A}^l, \boldsymbol{\theta}_\mathbf{H}^l\}$ is the set of trainable parameters.

**Optimality Module:** Due to the inexactness of these learning based architectures, the propagation module definitely brings errors when optimizing Eq. (1). To provide effective control for these iterative errors and generate strictly convergent propagations, we recall the designed prior $g$ and assume $\mathbf{x}_\mathcal{G}^l$ is one solution of Eq. (2) in the $l$-th stage, i.e.,

$$\mathbf{x}_\mathcal{G}^l \in \mathcal{G}(\tilde{\mathbf{x}}^l) := \arg\min_\mathbf{x} f(\mathbf{x}) + g(\mathbf{x}) + d(\mathbf{x}, \mathbf{x}_\mathcal{A}^l) - \langle \mathbf{x}, \boldsymbol{\epsilon}^l \rangle. \tag{4}$$

The error $\boldsymbol{\epsilon}^l$ has a specific form as $\boldsymbol{\epsilon}^l = \nabla f(\mathbf{x}_\mathcal{G}^l) + \nabla d(\mathbf{x}_\mathcal{G}^l, \mathbf{x}_\mathcal{A}^l) + \mathbf{u}_{\mathbf{x}_\mathcal{G}^l}$ by considering the first-order optimality condition of Eq. (4). Here $\mathbf{u}_{\mathbf{x}_\mathcal{G}^l} \in \partial g(\mathbf{x}_\mathcal{G}^l)$ is a limiting Ferchet subdifferential of $g$. Intuitively, it is necessary to introduce some criteria about $\boldsymbol{\epsilon}^l$ to illustrate the current propagation whether satisfied the desired convergence behavior. Fortunately, we can demonstrate that the convergence of our deep propagations can be successfully guaranteed by the following optimality error:

$$\|\psi(\boldsymbol{\epsilon}^l)\| \leq c^l \|\mathbf{x}_\mathcal{G}^l - \mathbf{x}^{l-1}\|. \tag{5}$$

Here $\psi(\boldsymbol{\epsilon}^l) = \boldsymbol{\epsilon}^l + \mu^l(\mathbf{x}_\mathcal{G}^l - \mathbf{x}^{l-1})/2 - \mathbf{H}(\mathbf{x}^{l-1} + \mathbf{x}_\mathcal{G}^l - 2\mathbf{x}_\mathcal{A}^l)$ is the error function and $c^l$ is a positive constant to reveal our tolerance of the inexactness at the $l$-th stage.

Therefore, as stated in the following Eq. (6), we adopt $\mathbf{x}_\mathcal{G}^l$ as the output of our optimality module in the $l$-th stage if the criterion in Eq. (5) is satisfied. Otherwise, we return to the previous stage and adopt a standard proximal gradient updating (i.e., feedback) to correct the propagation.

$$\mathcal{O}(\tilde{\mathbf{x}}^l, \mathbf{x}^{l-1}; \gamma^l) := \begin{cases} \mathcal{G}(\tilde{\mathbf{x}}^l) & \text{if Eq. (5) is satisfied,} \\ \mathrm{prox}_{\gamma^l g}(\mathbf{x}^{l-1} - \gamma^l(\nabla f(\mathbf{x}^{l-1}))) & \text{otherwise.} \end{cases} \tag{6}$$

In this way, our optimality module actually provides a mechanism with proximal operator to guide the propagations toward convergence.

Notice that both $\mathbf{x}_\mathcal{G}^l$ and $\boldsymbol{\epsilon}^l$ are abstracted in above optimality module. Actually, temporarily ignoring the learned prior and error in Eq (2), we can provide a practical calculative form of $\mathbf{x}_\mathcal{G}^l$ by calculating the traditionally designed prior appeared in Eq. (2) (i.e., Eq. (1), only with fidelity and designed priors) with a momentum proximal mechanism as follows,

$$\mathbf{x}_\mathcal{G}^l \in \mathrm{prox}_{\gamma^l g}\left( \tilde{\mathbf{x}}^l - \gamma^l\left(\nabla f(\tilde{\mathbf{x}}^l) + \mu^l(\tilde{\mathbf{x}}^l - \mathbf{x}^{l-1})\right) \right), \tag{7}$$

where $\mu^l$ is the trade-off parameter, and $\gamma^l$ denotes the step size. On the other hand, the updating of $\mathbf{x}_\mathcal{G}^l$ can also be reformulated[4] as $\mathbf{x}_\mathcal{G}^l \in \mathrm{prox}_{\gamma^l g}\left( \mathbf{x}_\mathcal{G}^l - \gamma^l\left(\nabla f(\mathbf{x}_\mathcal{G}^l) + \mu^l(\mathbf{x}_\mathcal{G}^l - \mathbf{x}_\mathcal{A}^l)\right) + \gamma^l \boldsymbol{\epsilon}^l \right)$. Thus, Combining it with Eq. (7), we can obtain a practical computable formulation of error function $\psi(\boldsymbol{\epsilon}^l)$ appeared in optimality error as

$$\psi(\boldsymbol{\epsilon}^l) = \frac{1}{\gamma^l}(\tilde{\mathbf{x}}^l - \mathbf{x}_\mathcal{G}^l) - \frac{\mu^l}{2}(2\tilde{\mathbf{x}}^l - \mathbf{x}_\mathcal{G}^l - \mathbf{x}^{l-1}) + \mathbf{H}(\mathbf{x}_\mathcal{G}^l - \mathbf{x}^{l-1}) + \nabla f(\mathbf{x}_\mathcal{G}^l) - \nabla f(\tilde{\mathbf{x}}^l).$$

# 4 Propagation and Optimization based Deep Model

Based on the above building-block modules, it is ready to introduce Propagation and Optimization based Deep Model (PODM). We first show how to apply PODM to perform fast and accurate model optimization and analyze its convergence behaviors in nonconvex and nonsmooth scenarios. Then we discuss how to establish end-to-end type PODM with relaxed optimality error to perform practical ensemble learning for challenging real-world applications.

## 4.1 PODM: A Deeply Trained Nonconvex Solver with Strict Convergence Guarantee

We demonstrate how to apply PODM for fast and accurate nonconvex optimization. It should be emphasized that different from most existing trainable iteration methods, which either incorporate networks into the iterations in heuristic manner (e.g., [33, 7]) or directly estimate data-dependent descent directions using networks (e.g., [17, 2]), PODM provides a nice mechanism with optimality error to control the training based propagations. It will be stated in the following that the main advantage of our PODM is that the convergence of our iterations can be strictly guaranteed, while no theoretical guarantees are provided for the above mentioned experientially trained iterations.

**PODM for Nonconvex and Nonsmooth Optimization:** We first illustrate the mechanism of PODM in Fig. 1. It can be seen that PODM consists of two fundamental modules, i.e., experientially designed (trainable) propagation module $\mathcal{P}$ and theoretically designed optimality module $\mathcal{O}$. It should be pointed out that to guarantee the theoretical convergence, only the parameters $\boldsymbol{\theta}_{\mathcal{A}}^l$ are learned when considering PODM as an accurate numerical solver[5].

**Convergence Behaviors Analysis:** Before providing our main theory, we give some statements about functions appeared in our optimization model. Specifically, we assume that $f$ is Lipschitz smooth, $g$ is proper and lower semi-continuous, $d$ is differential, and $\Phi$ is coercive[6]. All of these assumptions are fairly loose in most model optimization tasks.

**Proposition 1.** *Suppose that the optimality error in Eq.(5) (i.e., $\|\psi(\epsilon^l)\| \leq c^l \|\mathbf{x}_{\mathcal{G}}^l - \mathbf{x}^{l-1}\|$) is satisfied, then we have $\Phi(\mathbf{x}_{\mathcal{G}}^l) \leq \Phi(\mathbf{x}^{l-1}) - \left(\mu^l/4 - c^{l2}/\mu^l\right) \|\mathbf{x}_{\mathcal{G}}^l - \mathbf{x}^{l-1}\|^2$. In contrast, if the inequality in Eq. (5) is not satisfied and thus the variable is updated by $\mathbf{x}^l = \text{prox}_{\gamma^l g}(\mathbf{x}^{l-1} - \gamma^l(\nabla f(\mathbf{x}^{l-1})))$. Then we have $\Phi(\mathbf{x}^l) \leq \Phi(\mathbf{x}^{l-1}) - \left(1/(2\gamma^l) - L_f/2\right) \|\mathbf{x}^l - \mathbf{x}^{l-1}\|^2$, where $L_f$ is the Lipschitz modulus of $\nabla f(\mathbf{x})$.*

Actually, Proposition 1 provides us a nice descent property for PODM on the variational energy $\Phi(\mathbf{x})$ during iterations. That is, it is easy to obtain a non-increase sequence $\{\Phi(\mathbf{x}^l)\}_{l \in \mathbb{N}}$, which results in a limited value $\Phi^*$ so that $\lim_{l \to \infty} \Phi(\mathbf{x}^l) = \Phi^* < \infty$. Moreover, if $\{\mathbf{x}^l\}_{l \in \mathbb{N}}$ is bounded, there exists a convergence subsequence such that $\lim_{j \to \infty} \mathbf{x}^{l_j} = \mathbf{x}^*$, where $\{l_j\} \subset \{l\} \subset \mathbb{N}$. Then from the conclusions in Proposition 1, we also have the sum of $\|\mathbf{x}^l - \mathbf{x}^{l-1}\|^2$ from $l = 1$ to $l \to \infty$ is bounded. Thus we can further prove the following proposition.

**Proposition 2.** *Suppose $\mathbf{x}^*$ is any accumulation point of sequence $\{\mathbf{x}^l\}_{l \in \mathbb{N}}$ generalized by PODM, then there exists a subsequence $\{\mathbf{x}^{l_j}\}_{j \in \mathbb{N}}$ such that $\lim_{j \to \infty} \mathbf{x}^{l_j} = \mathbf{x}^*$, and $\lim_{j \to \infty} \Phi(\mathbf{x}^{l_j}) = \Phi(\mathbf{x}^*)$.*

Based on the above results, it is ready to establish the convergence results for our PODM when considering it as a numerical solver for nonconvex model optimization.

**Theorem 1.** *(Converge to the Critical Point of Eq.* (1)*) Suppose $f$ is proper and Lipschitz smooth, $g$ is proper and lower semi-continuous, and $\Phi$ is coercive. Then the output of PODM (i.e., $\{\mathbf{x}^l\}_{l \in \mathbb{N}}$) satisfies: 1. The limit points of $\{\mathbf{x}^l\}_{l \in \mathbb{N}}$ (denoted as $\Omega$) is a compact set; 2. All elements of $\Omega$ are the critical points of $\Phi$; 3. If $\Phi$ is a semi-algebraic function, $\{\mathbf{x}^l\}_{l \in \mathbb{N}}$ converges to a critical point of $\Phi$.*

In summary, we actually prove that PODM provides a novel strategy to iteratively guide the propagations of deep networks toward the critical point of the given nonconvex optimization model, leading to a fast and accurate numerical solver.

### 4.2 Relaxed PODM: An End-to-end Collaborative Learning Framework

It is shown in the above subsection that by enforcing a carefully designed optimality error and greedily train the networks, we can obtain a theoretically convergent solver for nonconvex optimization. However, it is indeed challenging to utilize strict mathematical models to exactly formulate the complex data distributions in real-world applications. Therefore, in this subsection, we would like to relax the theoretical constraint and develop a novel end-to-end learning framework to address real-world tasks. In particular, rather than only training the parameters $\theta_{\mathcal{A}}^l$ in given network $\mathcal{A}$, we also introduce flexible networks to learn parameters $\theta_{\mathbf{H}}^l$ in $\mathcal{F}$ at each layer. Therefore, at the $l$-th layer, we actually have two groups of learnable parameters, including $\theta_{\mathcal{A}}^l$ for $\mathcal{A}$ and $\theta_{\mathbf{H}}^l$ for $\mathcal{F}$. The forward propagation of the so-called Relaxed PODM (RPODM) at each stage can be summarized as $\mathbf{x}^l = \mathcal{G}\left(\mathcal{F}\left(\mathcal{A}\left(\mathbf{x}^{l-1}; \theta_{\mathcal{A}}^l\right); \theta_{\mathbf{H}}^l\right)\right)$. We would like to argue that RPODM actually provides a way to train the network structure using both domain knowledges and training data, thus results in a nice collaborative learning framework.

## 5 Experimental Results

We first analyze the convergence behaviors of PODM by applying it to solve the widely used nonconvex $\ell_p$-regularized sparse coding problem. Then, we evaluate the performance of our Relaxed PODM on the practical image restoration applications. All the experiments are conducted on a PC with Intel Core i7 CPU @ 3.6 GHz, 32 GB RAM and an NVIDIA GeForce GTX 1060 GPU.

### 5.1 PODM for $\ell_p$-regularized Sparse Coding

Now we consider the nonconvex $\ell_p$-regularized ($0 < p < 1$) sparse coding model: $\min_{\boldsymbol{\alpha}} \|\mathbf{D}\boldsymbol{\alpha} - \mathbf{o}\|^2 + \lambda\|\boldsymbol{\alpha}\|_p^p$, which has been widely used for synthetic image modeling [18, 22], subspace clustering [21] and motion segmentation [31], etc. Here $\lambda$ is the regularization parameter, $\mathbf{o}$, $\mathbf{D}$ and $\boldsymbol{\alpha}$ denote the observed signal, a given dictionary, and the corresponding sparse codes, respectively. In our experiments, we formulate $\mathbf{D}$ as the multiplication of the down-sampling operator and the inverse of a three-stage Haar wavelet transform [3], which results in a sparse coding based single image super-resolution formulation. We consider $\ell_{0.8}$-norm to enforce the sparsity constraint. As for PODM, we define $\mathbf{H} = \mu\mathbf{I}/2$ with $\mu = 1e - 2$ in the distance function $d$ (i.e., $d(\mathbf{x}, \mathbf{x}_{\mathcal{A}}) = \mu\|\mathbf{x} - \mathbf{x}_{\mathcal{A}}\|^2/2$) to establish the propagation and optimality modules. For fair comparison, we just adopt the network architecture used in existing works (i.e., IRCNN [33]) as $\mathcal{A}$ for PODM.

To verify the convergence properties of our framework, we plotted the iteration behaviors of PODM on example images from the commonly used "*Set5*" super-resolution benchmark [4] and compared it with the most popular numerical solvers (e.g., FISTA [3]) and the recently proposed representative network based iteration methods (e.g., IRCNN [33]). Fig. 2 showed the curves of relative error (i.e., $\log_{10}(\|\mathbf{x}^{l+1} - \mathbf{x}^l\|/\|\mathbf{x}^l\|)$), reconstruction error (i.e., $\|\mathbf{x}^l - \mathbf{x}_{\mathrm{gt}}\|/\|\mathbf{x}_{\mathrm{gt}}\|$), structural similarity (SSIM) and our optimality error defined in Eq. (5). It can be observed that the curves of relative error (i.e. subfigure (a)) for IRCNN is always oscillating and cannot converge even after 200 stages. This is mainly due to its naive network embedding strategy. Although with a little bit smooth relative errors, FISTA is much slower than our PODM. Meanwhile, we observed that PODM also has better performance than other two schemes in terms of the restoration error (in subfigure (b), lower is better) and SSIM (in subfigure (c), higher is better). Furthermore, we also explored the optimality error of our PODM. It can be seen from subfigure (d) that the optimality error is always satisfied. This means that the learnable architectures can always be used to improve the iteration process of PODM.

We then reported the average quantitative scores (i.e., PSNR and SSIM) on two benchmark datasets [4, 32]. As shown in Table 1, the quantitative performance of PODM are much better than all the compared methods on all up-sampling scales (i.e., $\times 2$, $\times 3$, $\times 4$).

### 5.2 Relaxed PODM for Image Restoration

Image restoration is one of the most challenging low-level vision problems, which aims to recover a latent clear image $\mathbf{u}$ from the blurred and noised observation $\mathbf{o}$. To evaluate the Relaxed PODM (RPODM) paradigm, we would like to formulate the image restoration task as $\mathbf{u}$ by solving $\min_{\mathbf{u}} \|\mathbf{k} \otimes \mathbf{u} - \mathbf{o}\|^2 + \chi_\Omega(\mathbf{u})$, where $\mathbf{k}$ and $\mathbf{n}$ are respectively the blur kernel and the noises,

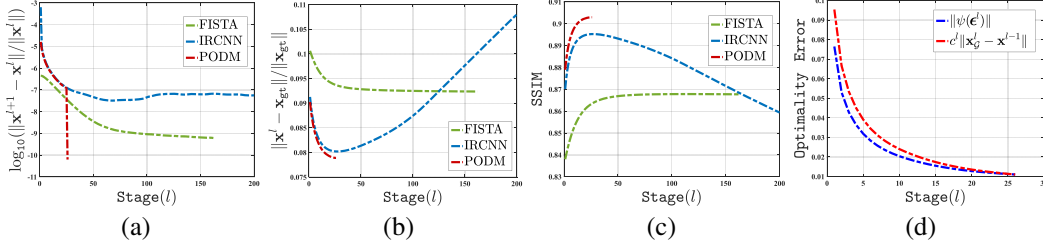

Figure 2: Convergence curves of FISTA, IRCNN, and our PODM. Subfigures (a)-(c) are the relative error, reconstruction error and SSIM, respectively. Subfigure (d) plots the "Optimality Error" appeared in PODM.

Table 1: Averaged quantitative performance on super-resolution with different up-sampling scales.

| Scale | Metric | [4] | | | [32] | | |
|---|---|---|---|---|---|---|---|
| | | FISTA | IRCNN | Ours | FISTA | IRCNN | Ours |
| ×2 | PSNR | 35.14 | 37.43 | **37.46** | 31.41 | 32.88 | **33.06** |
| | SSIM | 0.94 | 0.96 | **0.98** | 0.90 | 0.91 | **0.95** |
| ×3 | PSNR | 31.35 | 33.39 | **33.44** | 28.39 | 29.61 | **29.77** |
| | SSIM | 0.88 | 0.92 | **0.96** | 0.81 | 0.83 | **0.90** |
| ×4 | PSNR | 29.26 | 31.02 | **31.05** | 26.93 | 27.72 | **27.86** |
| | SSIM | 0.83 | 0.88 | **0.93** | 0.76 | 0.76 | **0.85** |

Table 2: Averaged quantitative performance on image restoration.

| | Metric | TV | HL | EPLL | CSF | RTF | MLP | IRCNN | Ours |
|---|---|---|---|---|---|---|---|---|---|
| [15] | PSNR | 29.38 | 30.12 | 31.65 | 32.74 | 33.26 | 31.32 | 32.51 | **34.06** |
| | SSIM | 0.88 | 0.90 | 0.93 | 0.93 | 0.94 | 0.90 | 0.92 | **0.97** |
| | TIME | 1.22 | **0.10** | 70.32 | 0.12 | 26.63 | 0.49 | 2.85 | 1.46 |
| [28] | PSNR | 30.67 | 31.03 | 32.44 | 31.55 | 32.45 | 31.47 | 32.61 | **32.62** |
| | SSIM | 0.85 | 0.85 | 0.88 | 0.87 | **0.89** | 0.86 | **0.89** | 0.89 |
| | TIME | 6.38 | **0.49** | 721.98 | 0.50 | 240.98 | 4.59 | 16.67 | 1.95 |

and $\chi_\Omega$ is the indicator function of the set $\Omega$. Here we define $\Omega = \{\mathbf{u}|\|\mathbf{u}\|_0 \leq s, a \leq [\mathbf{u}]_i \leq b$, with $s > 0$, $a = \min_i\{[\mathbf{o}]_i\}, b = \max_i\{[\mathbf{o}]_i\}, i = 1, \cdots, n.\}$ to enforce our fundamental constraints on $\mathbf{u}$. It is easy to check that the proximal operator corresponding to $\chi_\Omega$ can be written as $\mathtt{prox}_{\chi_\Omega}(\mathbf{u}) := \mathtt{HardThre}(\mathtt{Proj}_{[a,b]}(\mathbf{u}))$, where $\mathtt{HardThre}(\cdot)$ and $\mathtt{Proj}_{[a,b]}(\cdot)$ are the hard thresholding and projection operators, respectively. Then we would like to introduce learnable architectures to build RPODM. Specifically, by considering the distance measure $d(\mathbf{x}, \mathbf{x}_\mathcal{A})$ in filtered space, we define $\mathbf{H} = \mu \sum_{n=1}^N \mathbf{f}_n^\top \mathbf{f}_n / 2$, where $\{\mathbf{f}_n\}$ denote the filtering operations. In this way, the propagation module can be directly obtained by solving Eq. (3) in closed form, i.e., $\mathbf{x}_\mathcal{F} = (\mathbf{K}^\top \mathbf{K} + \mu \sum_{n=1}^N \mathbf{F}_n^\top \mathbf{F}_n)^{-1}(\mathbf{K}^\top \mathbf{o} + \mu \sum_{n=1}^N \mathbf{F}_n^\top \mathbf{F}_n \mathbf{x}_\mathcal{A})$, where $\mathbf{K}$ and $\{\mathbf{F}_n\}$ are block-circulant matrices corresponding to convolutions. Inspired by [14], here we just introduce a multilayer perceptron and the DCT basis to output the parameter $\mu$ and construct the filters $\{\mathbf{f}_n\}$, respectively. Then we adopt a CNN architecture with 6 convolutional layers (the first 5 layers followed by ELU [6] activations) as $\mathcal{A}$ for our deep propagation.

**Comparisons with State-of-the-art Methods**: We compared RPODM with several state-of-the-art image restoration approaches, including traditional numerical methods (e.g. TV [16], HL [12]), learning based methods (e.g. EPLL [34], MLP [26]), and deep unrolling methods (e.g. CSF [25], RTF [24], IRCNN [33]). We first conducted experiments on the most widely used Levin *et al.*' benchmark [15], with 32 blurry images of size $255 \times 255$. We also evaluated all these compared methods on the more challenging Sun *et al.*' benchmark [28], which includes 640 blurry images with 1% Gaussian noises, sizes range from $620 \times 1024$ to $928 \times 1024$. Table 2 reported the quantitative results (i.e., PSNR, SSIM and TIME (in seconds)). It can be seen that our proposed method

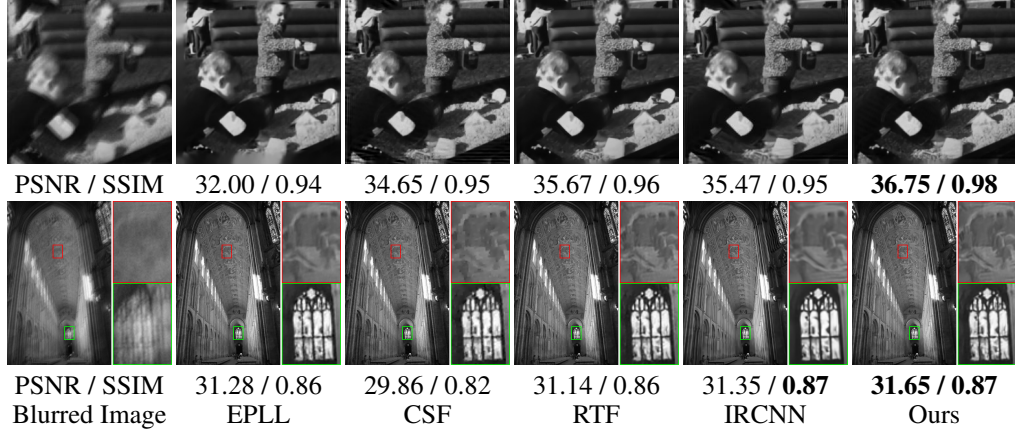

| PSNR / SSIM | 32.00 / 0.94 | 34.65 / 0.95 | 35.67 / 0.96 | 35.47 / 0.95 | **36.75 / 0.98** |

| PSNR / SSIM Blurred Image | 31.28 / 0.86 EPLL | 29.86 / 0.82 CSF | 31.14 / 0.86 RTF | 31.35 / **0.87** IRCNN | **31.65 / 0.87** Ours |

Figure 3: Image restoration results on two example images, where the inputs on the top and bottom rows are respectively from Levin *et al.*' and Sun *et al.*' benchmarks. The PSNR / SSIM are reported below each image.

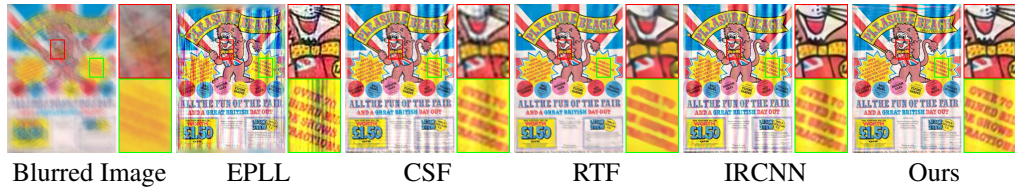

Figure 4: Image restoration results on the real blurry image.

can consistently obtain higher quantitative scores than other approaches, especially on Levin *et al.*' dataset [15]. As for the running time, we observed that RPODM is much faster than most conventional optimization based approaches and recently proposed learning based iteration methods (i.e., TV, EPLL, RTF, MLP and IRCNN). While the speeds of HL and CSF are slightly faster than RPODM. Unfortunately, the performance of these two simple methods are much worse than our approach. The qualitative comparisons in Fig. 3 also verified the effectiveness of RPODM.

**Real Blurry Images**: Finally, we evaluated RPODM on the real-world blurry images [11] (i.e., with unknown blur kernel and 1% additional Gaussian noises). We adopted the method in [23] to estimate a rough blur kernel. In Fig. 4, we compared the image restoration results of RPODM with other competitive methods (top 4 in Table 2, i.e., EPLL, CSF, RTF, and IRCNN) based on this estimated kernel. It can be seen that even with the roughly estimated kernel (maybe inexact), RPODM can still obtain clear image with richer details and more plentiful textures (see zoomed in regions).

## 6 Conclusions

This paper proposed Propagation and Optimization based Deep Model (PODM), a new paradigm to integrate principled domain knowledge and trainable architectures to build deep propagations for model optimization and machine learning. As a learning based numerical solver, we proved in theory that the sequences generated by PODM can successfully converge to the critical point of the given nonconvex and nonsmooth optimization model. Furthermore, by relaxing the optimality error, we actually also obtain a plug-and-play, collaborative, interpretable, and end-to-end deep model for real-world complex tasks. Extensive experimental results verified our theoretical investigations and demonstrated the effectiveness of the proposed framework.

## Acknowledgments

This work is partially supported by the National Natural Science Foundation of China (Nos. 61672125, 61733002, 61572096 and 61632019), and Fundamental Research Funds for the Central Universities.

## Footnotes

*Corresponding Author. Correspondence to `<rsliu@dlut.edu.cn>`.

[2]Here "globally" indicates that we generate a Cauchy sequence, thus the whole sequence is convergent.

[3]$d(\mathbf{x}, \mathbf{x}_\mathcal{A})$ actually is a special Bregman distance. $\mathbf{H}$ can be an assigned or learnable matrix. The distance $d(\mathbf{x}, \mathbf{x}_\mathcal{A}) = \mu\|\mathbf{x} - \mathbf{x}_\mathcal{A}\|^2$ if $\mathbf{H} = \mu\mathbf{I}$. We will detailed illustrate it on specific applications in experiments.

[4]The detained deductions on this equality reformulation can be found in the supplementary materials.

[5]Notice that both $\boldsymbol{\theta}_{\mathcal{A}}^l$ and $\boldsymbol{\theta}_{\mathbf{H}}^l$ are learnable in the Relaxed PODM, which will be introduced in Section 4.2. The algorithms of PODM and Relaxed PODM are presented in the supplementary materials.

[6]Due to the space limit, we omit the details of these definitions, all proofs of the following propositions and theorem. The detailed version is presented in the supplementary materials.

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
