[Supplementary Material · 2111-supplementary.pdf]

# A Bridging Framework for
# Model Optimization and Deep Propagation
# (Supplemental Materials)

**Risheng Liu[1,2]\*, Shichao Cheng[3], Xiaokun Liu[1], Long Ma[1], Xin Fan[1,2], Zhongxuan Luo[2,3]**
[1]International School of Information Science & Engineering, Dalian University of Technology
[2]Key Laboratory for Ubiquitous Network and Service Software of Liaoning Province
[3]School of Mathematical Science, Dalian University of Technology

## 1 Introduction

We first summarize the key equations and algorithms of Propagation and Optimization based Deep Model (PODM) in Section 2. Next, the practical calculable form of optimality error which appeared in optimality module is deduced in Section 3. Finally, necessary definitions and the proofs of PODM are provided in Section 4.

## 2 PODM for Nonconvex Optimization

We consider the following task-related variational model

$$\min_{\mathbf{x}} \Phi(\mathbf{x}) := f(\mathbf{x}) + g(\mathbf{x}), \tag{1}$$

where $f$ denotes the loss term and $g$ is related to the regularization term. Then the following quadratic penalized energy is proposed:

$$\min_{\mathbf{x}} \Phi(\mathbf{x}) + d(\mathbf{x}, \mathbf{x}_{\mathcal{A}}) - \langle \mathbf{x}, \boldsymbol{\epsilon} \rangle = \min_{\mathbf{x}} \overbrace{\underbrace{f(\mathbf{x})}_{\text{Fidelity}} + \underbrace{g(\mathbf{x})}_{\text{Designed prior}}}^{\text{Knowledge}} + \overbrace{\underbrace{d(\mathbf{x}, \mathbf{x}_{\mathcal{A}})}_{\text{Learned prior}} - \underbrace{\langle \mathbf{x}, \boldsymbol{\epsilon} \rangle}_{\text{Error}}}^{\text{Data}}. \tag{2}$$

where $\mathbf{x}_{\mathcal{A}} = \mathcal{A}(\mathbf{x}, \boldsymbol{\theta}_{\mathcal{A}})$ denotes the output of deep networks $\mathcal{A}$, $d(\mathbf{x}, \mathbf{x}_{\mathcal{A}})$ is the distance function which intents to introduce the output of network into the propagation module, and $\boldsymbol{\epsilon}$ denotes the error corresponding to Eq. (1).

We design propagation module by optimizing the following energy:

$$\mathbf{x}_{\mathcal{F}} = \mathcal{F}(\mathbf{x}_{\mathcal{A}}; \boldsymbol{\theta}_{\mathbf{H}}) := \arg\min_{\mathbf{x}} \{f(\mathbf{x}) + d(\mathbf{x}, \mathbf{x}_{\mathcal{A}})\}. \tag{3}$$

Thus, we formally formulate our propagation module $\mathcal{P}$ in the $l$-th stage with learnable parameters $\boldsymbol{\vartheta}^l = \{\boldsymbol{\theta}_{\mathcal{A}}^l, \boldsymbol{\theta}_{\mathbf{H}}^l\}$ as

$$\tilde{\mathbf{x}}^l = \mathcal{P}(\mathbf{x}^{l-1}; \boldsymbol{\vartheta}^l) := \mathcal{F}\left(\mathcal{A}\left(\mathbf{x}^{l-1}; \boldsymbol{\theta}_{\mathcal{A}}^l\right); \boldsymbol{\theta}_{\mathbf{H}}^l\right). \tag{4}$$

We also introduce the optimality module with proximal feedback to guarantee the theoretic convergence. We provide a practical calculable form of $\mathbf{x}_{\mathcal{G}}^l$ at the $l$-th stage as following:

$$\mathbf{x}_{\mathcal{G}}^l \in \mathcal{G}(\tilde{\mathbf{x}}^l; \gamma^l) := \mathtt{prox}_{\gamma^l g}(\tilde{\mathbf{x}}^l - \gamma^l(\nabla f(\tilde{\mathbf{x}}^l) + \mu^l(\tilde{\mathbf{x}}^l - \mathbf{x}^{l-1}))), \tag{5}$$

**Algorithm 1** Optimizing Eq. (1) via PODM.

**Require:** The input $\mathbf{x}^0$, the parameters $\boldsymbol{\vartheta}^l = \{\boldsymbol{\theta}_{\mathcal{A}}^l, \boldsymbol{\theta}_{\mathbf{H}}^l\}$, $c^l < \mu^l/2$, and $\gamma^l < 1/L_f$ with Lipschitz modulus $L_f$.

1: **while** not converged **do**
2:     // Propagation Module
3:     $\mathbf{x}_{\mathcal{A}}^l = \mathcal{A}(\mathbf{x}^{l-1}; \boldsymbol{\theta}_{\mathcal{A}}^l)$,
4:     $\tilde{\mathbf{x}}^l = \mathcal{F}(\mathbf{x}_{\mathcal{A}}^l; \boldsymbol{\theta}_{\mathbf{H}}^l)$,
5:     // Optimality Module
6:     $\mathbf{x}_{\mathcal{G}}^l \in \mathrm{prox}_{\gamma^l g}(\tilde{\mathbf{x}}^l - \gamma^l(\nabla f(\tilde{\mathbf{x}}^l) + \mu^l(\tilde{\mathbf{x}}^l - \mathbf{x}^{l-1})))$,
7:     **if** $\|\psi(\boldsymbol{\epsilon}^l)\| \le c^l \|\mathbf{x}_{\mathcal{G}}^l - \mathbf{x}^{l-1}\|$ **then**
8:         $\mathbf{x}^l = \mathbf{x}_{\mathcal{G}}^l$,
9:     **else**
10:       $\mathbf{x}^l \in \mathrm{prox}_{\gamma^l g}(\mathbf{x}^{l-1} - \gamma^l(\nabla f(\mathbf{x}^{l-1})))$.
11:    **end if**
12: **end while**

---

**Algorithm 2** Relaxed PODM.

**Require:** The input $\mathbf{x}^0$, the learnable parameters $\boldsymbol{\vartheta}^l = \{\boldsymbol{\theta}_{\mathcal{A}}^l, \boldsymbol{\theta}_{\mathbf{H}}^l\}$, and $\gamma^l < 1/L_f$ with Lipschitz modulus $L_f$.

1: **while** not converged **do**
2:     // Propagation Module
3:     $\mathbf{x}_{\mathcal{A}}^l = \mathcal{A}(\mathbf{x}^{l-1}; \boldsymbol{\theta}_{\mathcal{A}}^l)$,
4:     $\tilde{\mathbf{x}}^l = \mathcal{F}(\mathbf{x}_{\mathcal{A}}^l; \boldsymbol{\theta}_{\mathbf{H}}^l)$,
5:     // Optimality Module
6:     $\mathbf{x}^l \in \mathcal{G}(\tilde{\mathbf{x}}^l; \gamma^l)$.
7: **end while**

---

where $\gamma^l$ is the step size. In our assumption, $\mathbf{x}_{\mathcal{G}}^l$ is one of the optimal solutions of Eq. (2), we can deduce that

$$\boldsymbol{\epsilon}^l = \nabla f(\mathbf{x}_{\mathcal{G}}^l) + \nabla d(\mathbf{x}_{\mathcal{G}}^l, \mathbf{x}_{\mathcal{A}}^l) + \mathbf{u}_{\mathbf{x}_{\mathcal{G}}^l}, \text{ where } \mathbf{u}_{\mathbf{x}_{\mathcal{G}}^l} \in \partial g(\mathbf{x}_{\mathcal{G}}^l), \tag{6}$$

by calculating the first-order optimality condition. Then, we build the following optimality error:

$$\|\psi(\boldsymbol{\epsilon}^l)\| \le c^l \|\mathbf{x}_{\mathcal{G}}^l - \mathbf{x}^{l-1}\|, \tag{7}$$

where $\psi(\boldsymbol{\epsilon}^l) = \boldsymbol{\epsilon}^l + \mu^l(\mathbf{x}_{\mathcal{G}}^l - \mathbf{x}^{l-1})/2 - \mathbf{H}(\mathbf{x}^{l-1} + \mathbf{x}_{\mathcal{G}}^l - 2\mathbf{x}_{\mathcal{A}}^l)$ is the error function and $c^l$ is a positive constant to reveal our tolerance of the inexactness at the $l$-th stage. Finally, we define our optimality module by the judging mechanism with proximal feedback as following:

$$\mathcal{O}(\tilde{\mathbf{x}}^l, \mathbf{x}^{l-1}; \gamma^l) := \begin{cases} \mathcal{G}(\tilde{\mathbf{x}}^l) & \text{if Eq. (7) is satisfied,} \\ \mathrm{prox}_{\gamma^l g}(\mathbf{x}^{l-1} - \gamma^l(\nabla f(\mathbf{x}^{l-1}))) & \text{otherwise.} \end{cases} \tag{8}$$

Finally, we present the complete PODM based nonconvex optimization in Alg. 1. Relaxing the theoretical constraint, we develop a *plug-and-play*, *collaborative*, *interpretable*, and *end-to-end* deep learning framework RPODM as shown in Alg. 2.

## 3 Practical Calculable Form of Optimality Error

In this section, we offer practical calculable form of $\psi(\boldsymbol{\epsilon}^l)$ appeared in the optimality error Eq. (7). In fact, the sub-differential $\mathbf{u}_{\mathbf{x}_{\mathcal{G}}^l}$ included in $\boldsymbol{\epsilon}^l$ and $\psi(\boldsymbol{\epsilon}^l)$ is challenging to calculate, thus a calculable form of $\mathbf{u}_{\mathbf{x}_{\mathcal{G}}^l}$ or $\psi(\boldsymbol{\epsilon}^l)$ is necessary for executing our algorithm. To obtain the calculable form, we first consider the practical calculation of $\mathbf{x}_{\mathcal{G}}^l$ at the $l$-th stage, i.e.,

$$\mathbf{x}_{\mathcal{G}}^l \in \mathrm{prox}_{\gamma^l g}(\tilde{\mathbf{x}}^l - \gamma^l(\nabla f(\tilde{\mathbf{x}}^l) + \mu^l(\tilde{\mathbf{x}}^l - \mathbf{x}^{l-1}))). \tag{9}$$

On the other hand, from the definition of $\boldsymbol{\epsilon}^l$ in Eq. (6), we have

$$\mathbf{u}_{\mathbf{x}_{\mathcal{G}}^l} = \boldsymbol{\epsilon}^l - \nabla f(\mathbf{x}_{\mathcal{G}}^l) - \nabla d(\mathbf{x}_{\mathcal{G}}^l, \mathbf{x}_{\mathcal{A}}^l) = \boldsymbol{\epsilon}^l - \nabla f(\mathbf{x}_{\mathcal{G}}^l) - 2\mathbf{H}(\mathbf{x}_{\mathcal{G}}^l - \mathbf{x}_{\mathcal{A}}^l) \in \partial g(\mathbf{x}_{\mathcal{G}}^l).$$

By the property of proximal operation, we have

$$
\begin{aligned}
& 0 \in \gamma^l \left( \partial g(\mathbf{x}_{\mathcal{G}}^l) - \mathbf{u}_{\mathbf{x}_{\mathcal{G}}^l} \right) = \gamma^l \partial g(\mathbf{x}_{\mathcal{G}}^l) + \mathbf{x}_{\mathcal{G}}^l - \left( \mathbf{x}_{\mathcal{G}}^l + \gamma^l \mathbf{u}_{\mathbf{x}_{\mathcal{G}}^l} \right) \\
& \Leftrightarrow \mathbf{x}_{\mathcal{G}}^l \in \arg\min_{\mathbf{x}} g(\mathbf{x}) + \tfrac{1}{2\gamma^l} \| \mathbf{x} - \left( \mathbf{x}_{\mathcal{G}}^l + \gamma^l \mathbf{u}_{\mathbf{x}_{\mathcal{G}}^l} \right) \|^2 \\
& \Leftrightarrow \mathbf{x}_{\mathcal{G}}^l \in \mathtt{prox}_{\gamma^l g} \left( \mathbf{x}_{\mathcal{G}}^l + \gamma^l \mathbf{u}_{\mathbf{x}_{\mathcal{G}}^l} \right) \\
& \Leftrightarrow \mathbf{x}_{\mathcal{G}}^l \in \mathtt{prox}_{\gamma^l g} \left( \mathbf{x}_{\mathcal{G}}^l - \gamma^l (\nabla f(\mathbf{x}_{\mathcal{G}}^l) + 2\mathbf{H}(\mathbf{x}_{\mathcal{G}}^l - \mathbf{x}_{\mathcal{A}}^l)) + \gamma^l \boldsymbol{\epsilon}^l \right).
\end{aligned}
\tag{10}
$$

Combing Eqs. (9) with (10), we easily obtain the calculable form of $\psi(\boldsymbol{\epsilon}^l)$ as

$$
\psi(\boldsymbol{\epsilon}^l) = \frac{1}{\gamma^l} (\tilde{\mathbf{x}}^l - \mathbf{x}_{\mathcal{G}}^l) - \frac{\mu^l}{2} (2\tilde{\mathbf{x}}^l - \mathbf{x}_{\mathcal{G}}^l - \mathbf{x}^{l-1}) + \mathbf{H}(\mathbf{x}_{\mathcal{G}}^l - \mathbf{x}^{l-1}) + \nabla f(\mathbf{x}_{\mathcal{G}}^l) - \nabla f(\tilde{\mathbf{x}}^l).
$$

# 4  Proofs of PODM

Before proving our main theory, we first give some mathematical definitions, which will be necessary in our analysis.

**Definition 1.** *The following definitions are standard in variational analysis and optimization. Please also refer to [1, 2] for more detailed introductions.*

1. *A function $f : \mathbb{R}^n \to (-\infty, +\infty]$ is L-Lipschitz smooth if $f$ is differentiable and there exists $L > 0$ such that*

$$
\| \nabla f(\mathbf{x}) - \nabla f(\mathbf{y}) \| \le L \| \mathbf{x} - \mathbf{y} \|, \; \forall \, \mathbf{x}, \mathbf{y} \in \mathbb{R}^n.
$$

2. *$f$ is said to be proper and lower semi-continuous if its domain $dom(f) \ne \emptyset$, where $dom(f) := \{ \mathbf{x} \in \mathbb{R}^n : f(\mathbf{x}) < +\infty \}$ and $\liminf_{\mathbf{x} \to \mathbf{y}} f(\mathbf{x}) \ge f(\mathbf{y})$ at any point $\mathbf{y} \in dom(f)$.*

3. *$f$ is said to be coercive if $f$ is bounded from below and $f \to \infty$ if $\| \mathbf{x} \| \to \infty$, where $\| \cdot \|$ is the $\ell_2$-norm.*

4. *$f$ is called semi-algebraic[2] if its graph $\{ (\mathbf{x}, z) \in \mathbb{R}^{n+1} : f(\mathbf{x}) = z \}$ is a semi-algebraic set. Here we define the set $\mathcal{S} \subseteq \mathbb{R}^n$ as a semi-algebraic set if it satisfies $\mathcal{S} = \bigcup_{j=1}^p \bigcap_{i=1}^q \{ \mathbf{x} \in \mathbb{R}^n : r_{ij}(\mathbf{x}) = 0 \text{ and } h_{ij}(\mathbf{x}) < 0 \}$, where $r_{ij}, h_{ij} : \mathbb{R}^n \to \mathbb{R}$ are real polynomial functions.*

5. *A point $\mathbf{x}^*$ whose subdifferential $\partial f(\mathbf{x}^*)$ contains $\mathbf{0}$ are called the critical point of $f$.*

**Proposition 1.** *Suppose that the optimality error in Eq.(7) (i.e., $\| \psi(\boldsymbol{\epsilon}^l) \| \le c^l \| \mathbf{x}_{\mathcal{G}}^l - \mathbf{x}^{l-1} \|$) is satisfied, then we have $\Phi(\mathbf{x}_{\mathcal{G}}^l) \le \Phi(\mathbf{x}^{l-1}) - \left( \mu^l/4 - c^{l2}/\mu^l \right) \| \mathbf{x}_{\mathcal{G}}^l - \mathbf{x}^{l-1} \|^2$. In contrast, if the inequality in Eq. (7) is not satisfied and thus the variable is updated by $\mathbf{x}^l = \mathtt{prox}_{\gamma^l g}(\mathbf{x}^{l-1} - \gamma^l (\nabla f(\mathbf{x}^{l-1})))$. Then we have $\Phi(\mathbf{x}^l) \le \Phi(\mathbf{x}^{l-1}) - \left( 1/(2\gamma^l) - L_f/2 \right) \| \mathbf{x}^l - \mathbf{x}^{l-1} \|^2$, where $L_f$ is the Lipschitz modulus of $\nabla f(\mathbf{x})$.*

*Proof.* To illustrate the importance of our optimality error, we first prove the unequal relationship of $\Phi(\mathbf{x}_{\mathcal{G}}^l)$ and $\Phi(\mathbf{x}^{l-1})$. Considering the definition of distant function $d(\mathbf{x}, \mathbf{x}_{\mathcal{A}}^l)$, we have

$$
d(\mathbf{x}, \mathbf{x}_{\mathcal{A}}^l) = h(\mathbf{x}) - h(\mathbf{x}_{\mathcal{A}}^l) - \langle \nabla h(\mathbf{x}_{\mathcal{A}}^l), \mathbf{x} - \mathbf{x}_{\mathcal{A}}^l \rangle = \langle \mathbf{x} - \mathbf{x}_{\mathcal{A}}^l, \mathbf{H}(\mathbf{x} - \mathbf{x}_{\mathcal{A}}^l) \rangle.
$$

Thus, the quadratic penalized energy Eq. (2) in $l$-th stage can be rewrite as

$$
\begin{aligned}
\mathbf{x}_{\mathcal{G}}^l & \in \arg\min_{\mathbf{x}} g(\mathbf{x}) + f(\mathbf{x}) + d(\mathbf{x}, \mathbf{x}_{\mathcal{A}}^l) - \langle \mathbf{x}, \boldsymbol{\epsilon}^l \rangle \\
& = \arg\min_{\mathbf{x}} g(\mathbf{x}) + f(\mathbf{x}) + \langle \mathbf{x} - \mathbf{x}_{\mathcal{A}}^l, \mathbf{H}(\mathbf{x} - \mathbf{x}_{\mathcal{A}}^l) \rangle - \langle \mathbf{x}, \boldsymbol{\epsilon}^l \rangle \\
& = \arg\min_{\mathbf{x}} g(\mathbf{x}) + f(\mathbf{x}) + \langle \mathbf{x} - \mathbf{x}_{\mathcal{A}}^l, \mathbf{H}(\mathbf{x} - \mathbf{x}_{\mathcal{A}}^l) \rangle - \langle \mathbf{x} - \mathbf{x}^{l-1}, \boldsymbol{\epsilon}^l \rangle,
\end{aligned}
\tag{11}
$$

when $\mathbf{x}_\mathcal{G}^l$ is an exact solution of Eq. (2). Based on the above deduction, we easily obtain

$$
\begin{aligned}
&g(\mathbf{x}_\mathcal{G}^l) + f(\mathbf{x}_\mathcal{G}^l) + \langle \mathbf{x}_\mathcal{G}^l - \mathbf{x}_\mathcal{A}^l, \mathbf{H}(\mathbf{x}_\mathcal{G}^l - \mathbf{x}_\mathcal{A}^l)\rangle + \tfrac{\mu^l}{2}\|\mathbf{x}_\mathcal{G}^l - \mathbf{x}^{l-1}\|^2 \\
&\quad - \langle \mathbf{x}_\mathcal{G}^l - \mathbf{x}^{l-1}, \boldsymbol{\epsilon}^l + \tfrac{\mu^l}{2}(\mathbf{x}_\mathcal{G}^l - \mathbf{x}^{l-1})\rangle \\
&\leq g(\mathbf{x}^{l-1}) + f(\mathbf{x}^{l-1}) + \langle \mathbf{x}^{l-1} - \mathbf{x}_\mathcal{A}^l, \mathbf{H}(\mathbf{x}^{l-1} - \mathbf{x}_\mathcal{A}^l)\rangle \\
&\Rightarrow \Phi(\mathbf{x}_\mathcal{G}^l) \leq \Phi(\mathbf{x}^{l-1}) - \tfrac{\mu^l}{2}\|\mathbf{x}_\mathcal{G}^l - \mathbf{x}^{l-1}\|^2 + \langle \mathbf{x}^{l-1} - \mathbf{x}_\mathcal{G}^l, \mathbf{H}(\mathbf{x}^{l-1} + \mathbf{x}_\mathcal{G}^l - 2\mathbf{x}_\mathcal{A}^l)\rangle \\
&\quad + \langle \mathbf{x}_\mathcal{G}^l - \mathbf{x}^{l-1}, \boldsymbol{\epsilon}^l + \tfrac{\mu^l}{2}(\mathbf{x}_\mathcal{G}^l - \mathbf{x}^{l-1})\rangle \\
&\leq \Phi(\mathbf{x}^{l-1}) - \tfrac{\mu^l}{2}\|\mathbf{x}_\mathcal{G}^l - \mathbf{x}^{l-1}\|^2 + \langle \mathbf{x}_\mathcal{G}^l - \mathbf{x}^{l-1}, \boldsymbol{\epsilon}^l + \tfrac{\mu^l}{2}(\mathbf{x}_\mathcal{G}^l - \mathbf{x}^{l-1}) - \mathbf{H}(\mathbf{x}_\mathcal{G}^l + \mathbf{x}^{l-1} - 2\mathbf{x}_\mathcal{A}^l)\rangle \\
&= \Phi(\mathbf{x}^{l-1}) - \tfrac{\mu^l}{2}\|\mathbf{x}_\mathcal{G}^l - \mathbf{x}^{l-1}\|^2 + \langle \mathbf{x}_\mathcal{G}^l - \mathbf{x}^{l-1}, \psi(\boldsymbol{\epsilon}^l)\rangle \\
&\leq \Phi(\mathbf{x}^{l-1}) - \tfrac{\mu^l}{2}\|\mathbf{x}_\mathcal{G}^l - \mathbf{x}^{l-1}\|^2 + c^l\|\mathbf{x}_\mathcal{G}^l - \mathbf{x}^{l-1}\|^2 \\
&\leq \Phi(\mathbf{x}^{l-1}) - \left(\tfrac{\mu^l}{2} - c^l\right)\|\mathbf{x}_\mathcal{G}^l - \mathbf{x}^{l-1}\|^2,
\end{aligned}
\tag{12}
$$

in which the penultimate inequality holds under the designed optimality error in Eq. (7) is satisfied. Thus we prove that

$$
\Phi(\mathbf{x}_\mathcal{G}^l) \leq \Phi(\mathbf{x}^{l-1}) - \left(\frac{\mu^l}{2} - c^l\right)\|\mathbf{x}_\mathcal{G}^l - \mathbf{x}^{l-1}\|^2,
$$

and can deduce that $\Phi(\mathbf{x}_\mathcal{G}^l) \leq \Phi(\mathbf{x}^{l-1})$ when $\mu^l > 2c^l$.

Then we derive the relationship of $\Phi(\mathbf{x}^l)$ and $\Phi(\mathbf{x}^{l-1})$ under Eq. (7) is fail. From the optimization module in Eq. (8), we have

$$
\begin{aligned}
\mathbf{x}^l &\in \text{prox}_{\gamma^l g}(\mathbf{x}^{l-1} - \gamma^l(\nabla f(\mathbf{x}^{l-1}))) \\
&= \arg\min_{\mathbf{x}} g(\mathbf{x}) + \langle \nabla f(\mathbf{x}^{l-1}), \mathbf{x} - \mathbf{x}^{l-1}\rangle + \tfrac{1}{2\gamma^l}\|\mathbf{x} - \mathbf{x}^{l-1}\|^2,
\end{aligned}
\tag{13}
$$

which means that

$$
g(\mathbf{x}^l) + \langle \nabla f(\mathbf{x}^{l-1}), \mathbf{x}^l - \mathbf{x}^{l-1}\rangle + \tfrac{1}{2\gamma^l}\|\mathbf{x}^l - \mathbf{x}^{l-1}\|^2 \leq g(\mathbf{x}^{l-1}).
\tag{14}
$$

Moreover, we also have

$$
f(\mathbf{x}^l) \leq f(\mathbf{x}^{l-1}) + \langle \nabla f(\mathbf{x}^{l-1}), \mathbf{x}^l - \mathbf{x}^{l-1}\rangle + \tfrac{L_f}{2}\|\mathbf{x}^l - \mathbf{x}^{l-1}\|^2,
\tag{15}
$$

since of $f$ is Lipschitz smooth and $L_f$ is the Lipschitz moduli of $\nabla f$. Combining Eqs. (14) and (15), we can obviously obtain

$$
\Phi(\mathbf{x}^l) \leq \Phi(\mathbf{x}^{l-1}) - \left(\tfrac{1}{2\gamma^l} - \tfrac{L_f}{2}\right)\|\mathbf{x}^l - \mathbf{x}^{l-1}\|^2.
\tag{16}
$$

When $\gamma^l < 1/L_f$ is established, we have $\Phi(\mathbf{x}^l) \leq \Phi(\mathbf{x}^{l-1})$.

By now, we deduce all conclusions in this Proposition.

$\square$

**Proposition 2.** *Suppose $\mathbf{x}^*$ is any accumulation point of sequence $\{\mathbf{x}^l\}_{l\in\mathbb{N}}$ generalized by PODM, then there exists a subsequence $\{\mathbf{x}^{l_j}\}_{j\in\mathbb{N}}$ such that $\lim_{j\to\infty} \mathbf{x}^{l_j} = \mathbf{x}^*$, and $\lim_{j\to\infty} \Phi(\mathbf{x}^{l_j}) = \Phi(\mathbf{x}^*)$.*

*Proof.* From Proposition 1, we have $\Phi(\mathbf{x}^l) = \Phi(\mathbf{x}_\mathcal{G}^l) \leq \Phi(\mathbf{x}^{l-1})$ when both optimality error and $\mu^l > 2c^l$ are satisfied. Otherwise, we directly have the relationship of the variational energy $\Phi(\mathbf{x}^l) \leq \Phi(\mathbf{x}^{l-1})$ while $\gamma^l < 1/L_f$. Thus we find that $\{\Phi(\mathbf{x}^l)\}_{l\in\mathbb{N}}$ is a non-increasing sequence, i.e., for any $l \in \mathbb{N}$,

$$
\Phi(\mathbf{x}^l) \leq \Phi(\mathbf{x}^{l-1}) \leq \dots \Phi(\mathbf{x}^0).
$$

Since both $f$ and $g$ are proper, we have $\Phi(\mathbf{x}^l) \geq \inf \Phi > -\infty$ and deduce $\{\Phi(\mathbf{x}^l)\}_{l\in\mathbb{N}}$ is bounded. Thus $\{\Phi(\mathbf{x}^l)\}_{l\in\mathbb{N}}$ is a convergent sequence, i.e.,

$$
\lim_{l\to\infty} \Phi(\mathbf{x}^l) = \Phi^*,
\tag{17}
$$

which $\Phi^*$ is the limit value. Furthermore, we also deduce that the variable sequence $\{\mathbf{x}^l\}_{l\in\mathbb{N}}$ is bounded and have accumulation points since $\Phi$ is coercive. Assuming $\Omega$ is the set of accumulation points, there exists a subsequence $\{l_j\} \subset \mathbb{N}$ such that $\lim_{j\to\infty} \mathbf{x}^{l_j} = \mathbf{x}^*$ for any $\mathbf{x}^* \in \Omega$.

Next, we will prove $\lim_{l \to \infty} \Phi(\mathbf{x}^{l_j}) = \Phi(\mathbf{x}^*)$. Following the calculation of $\mathbf{x}_{\mathcal{G}}^l$ in Eq. (11), we have

$$
\begin{aligned}
& g(\mathbf{x}^l) + f(\mathbf{x}^l) + \langle \mathbf{x}^l - \mathbf{x}_{\mathcal{A}}^l, \mathbf{H}(\mathbf{x}^l - \mathbf{x}_{\mathcal{A}}^l) \rangle - \langle \mathbf{x}^l, \boldsymbol{\epsilon}^l \rangle \\
& \leq g(\mathbf{x}^*) + f(\mathbf{x}^*) + \langle \mathbf{x}^* - \mathbf{x}_{\mathcal{A}}^l, \mathbf{H}(\mathbf{x}^* - \mathbf{x}_{\mathcal{A}}^l) \rangle - \langle \mathbf{x}^*, \boldsymbol{\epsilon}^l \rangle \\
& \Rightarrow g(\mathbf{x}^l) \leq g(\mathbf{x}^*) + f(\mathbf{x}^*) - f(\mathbf{x}^l) - \langle \mathbf{x}^* - \mathbf{x}^l, \boldsymbol{\epsilon}^l - \mathbf{H}(\mathbf{x}^* + \mathbf{x}^l - 2\mathbf{x}_{\mathcal{A}}^l) \rangle,
\end{aligned}
\tag{18}
$$

when the optimality error is satisfied. Otherwise, following Eq. (13), we have

$$
\begin{aligned}
& g(\mathbf{x}^l) + \langle \nabla f(\mathbf{x}^{l-1}), \mathbf{x}^l - \mathbf{x}^{l-1} \rangle + \tfrac{1}{2\gamma^l} \|\mathbf{x}^l - \mathbf{x}^{l-1}\|^2 \\
& \leq g(\mathbf{x}^*) + \langle \nabla f(\mathbf{x}^{l-1}), \mathbf{x}^* - \mathbf{x}^{l-1} \rangle + \tfrac{1}{2\gamma^l} \|\mathbf{x}^* - \mathbf{x}^{l-1}\|^2.
\end{aligned}
\tag{19}
$$

Define $\mathbb{N}_1 = \{l \mid \|\psi(\boldsymbol{\epsilon}^l)\| \leq c^l \|\mathbf{x}_{\mathcal{G}}^l - \mathbf{x}^{l-1}\|\}$, and $\mathbb{N}_2 = \{l \mid \mathbb{N} - \mathbb{N}_1\}$, we have

$$
\begin{cases}
g(\mathbf{x}^l) \leq g(\mathbf{x}^*) + f(\mathbf{x}^*) - f(\mathbf{x}^l) - \langle \mathbf{x}^* - \mathbf{x}^l, \boldsymbol{\epsilon}^l - \mathbf{H}(\mathbf{x}^* + \mathbf{x}^l - 2\mathbf{x}_{\mathcal{A}}^l) \rangle & \text{if } l \in \mathbb{N}_1, \\
g(\mathbf{x}^l) \leq g(\mathbf{x}^*) + \langle \nabla f(\mathbf{x}^{l-1}), \mathbf{x}^* - \mathbf{x}^l \rangle + \tfrac{1}{2\gamma^l} \langle \mathbf{x}^* - \mathbf{x}^l, \mathbf{x}^* + \mathbf{x}^l - 2\mathbf{x}^{l-1} \rangle & \text{otherwise.}
\end{cases}
\tag{20}
$$

By taking $\lim \sup$ on the above inequality, we deduce $\lim_{j \to \infty} \sup g(\mathbf{x}^{l_j}) \leq g(\mathbf{x}^*)$ when $l = l_j$. On the other hand, since $g$ is lower semi-continuous and $\mathbf{x}^{l_j} \to \mathbf{x}^*$, it follows that $\lim_{j \to \infty} \inf g(\mathbf{x}^{l_j}) \geq g(\mathbf{x}^*)$. Thus, we have $\lim_{j \to \infty} g(\mathbf{x}^{l_j}) = g(\mathbf{x}^*)$. Considering the continuity of $f$ yields $\lim_{j \to \infty} f(\mathbf{x}^{l_j}) = f(\mathbf{x}^*)$, thus we conclude

$$
\lim_{j \to \infty} \Phi(\mathbf{x}^{l_j}) = \Phi(\mathbf{x}^*).
\tag{21}
$$

$\square$

**Theorem 1.** *(Converge to the Critical Point of Eq.* (1)*) Suppose $f$ is proper and Lipschitz smooth, $g$ is proper and lower semi-continuous, and $\Phi$ is coercive. Then the output of PODM (i.e., $\{\mathbf{x}^l\}_{l \in \mathbb{N}}$) satisfies: 1. The limit points of $\{\mathbf{x}^l\}_{l \in \mathbb{N}}$ (denoted as $\Omega$) is a compact set; 2. All elements of $\Omega$ are the critical points of $\Phi$; 3. If $\Phi$ is a semi-algebraic function, $\{\mathbf{x}^l\}_{l \in \mathbb{N}}$ converges to a critical point of $\Phi$.*

*Proof.* It is obvious that $\Omega$ is a compact set since the limit points of $\{\mathbf{x}^l\}_{l \in \mathbb{N}}$ is closed and bounded from the proof of Proposition 2.

Next, we will prove that all elements of $\Omega$ are the critical points of $\Phi$. Recalled the proof of Propositions 1 and 2, we first find that $\lim_{j \to \infty} \Phi(\mathbf{x}^*) = \Phi^*$ for any $\mathbf{x}^* \in \Omega$ from Eqs. (17) and (21). We also find that

$$
\begin{cases}
\left( \tfrac{\mu^l}{2} - c^l \right) \|\mathbf{x}^l - \mathbf{x}^{l-1}\|^2 \leq \Phi(\mathbf{x}^{l-1}) - \Phi(\mathbf{x}^l) & \text{if } l \in \mathbb{N}_1, \\
\left( \tfrac{1}{2\gamma^l} - \tfrac{L_f}{2} \right) \|\mathbf{x}^l - \mathbf{x}^{l-1}\|^2 \leq \Phi(\mathbf{x}^{l-1}) - \Phi(\mathbf{x}^l) & \text{otherwise.}
\end{cases}
\tag{22}
$$

Summing over $l$, we further have

$$
\begin{aligned}
& \min_{l \in \mathbb{N}} \{ \tfrac{\mu^l}{2} - c^l, \tfrac{1}{2\gamma^l} - \tfrac{L_f}{2} \} \sum_{l=1}^{\infty} \|\mathbf{x}^l - \mathbf{x}^{l-1}\|^2 \\
& \leq \min_{l \in \mathbb{N}_1} \{ \tfrac{\mu^l}{2} - c^l \} \sum_{l \in \mathbb{N}_1} \|\mathbf{x}^l - \mathbf{x}^{l-1}\|^2 + \min_{l \in \mathbb{N}_2} \{ \tfrac{1}{2\gamma^l} - \tfrac{L_f}{2} \} \sum_{l \in \mathbb{N}_2} \|\mathbf{x}^l - \mathbf{x}^{l-1}\|^2 \\
& \leq \sum_{l \in \mathbb{N}_1} (\tfrac{\mu^l}{2} - c^l) \|\mathbf{x}^l - \mathbf{x}^{l-1}\|^2 + \sum_{l \in \mathbb{N}_2} (\tfrac{1}{2\gamma^l} - \tfrac{L_f}{2}) \|\mathbf{x}^l - \mathbf{x}^{l-1}\|^2 \\
& \leq \sum_{l \in \mathbb{N}_1} \left( \Phi(\mathbf{x}^{l-1}) - \Phi(\mathbf{x}^l) \right) + \sum_{l \in \mathbb{N}_2} \left( \Phi(\mathbf{x}^{l-1}) - \Phi(\mathbf{x}^l) \right) \\
& = \sum_{l=1}^{\infty} \left( \Phi(\mathbf{x}^{l-1}) - \Phi(\mathbf{x}^l) \right) = \Phi(\mathbf{x}^0) - \Phi^*.
\end{aligned}
\tag{23}
$$

The above inequality implies $\|\mathbf{x}^l - \mathbf{x}^{l-1}\| \to 0$ when $l \to \infty$. Considering the sub-differential $\partial \Phi$, we have

$$
\begin{aligned}
\|\partial \Phi(\mathbf{x}^l)\| &= \|\boldsymbol{\epsilon}^l - \nabla d(\mathbf{x}^l, \mathbf{x}_{\mathcal{A}}^l)\| \\
&= \|\psi(\boldsymbol{\epsilon}^l) - \tfrac{\mu^l}{2}(\mathbf{x}^l - \mathbf{x}^{l-1}) - \mathbf{H}(\mathbf{x}^l - \mathbf{x}^{l-1})\| \\
&\leq (c^l + \tfrac{\mu^l}{2} + \rho_{\mathbf{H}}) \|\mathbf{x}^l - \mathbf{x}^{l-1}\|.
\end{aligned}
\tag{24}
$$

under the optimality error in Eq. (7) holds. Here, $\rho_{\mathbf{H}}$ is the spectral norm of $\mathbf{H}$. Otherwise, from Eq. (13), we deduce that

$$\|\partial \Phi(\mathbf{x}^l)\| = \|\nabla f(\mathbf{x}^l) - \nabla f(\mathbf{x}^{l-1}) - \frac{1}{\gamma^l}(\mathbf{x}^l - \mathbf{x}^{l-1})\|. \tag{25}$$

Eqs. (24) and (25) imply that

$$\begin{cases} \|\partial \Phi(\mathbf{x}^l)\| \leq (c^l + \frac{\mu^l}{2} + \rho_{\mathbf{H}})\|\mathbf{x}^l - \mathbf{x}^{l-1}\| & \text{if } l \in \mathbb{N}_1, \\ \|\partial \Phi(\mathbf{x}^l)\| \leq (L_f + \frac{1}{\gamma^l})\|\mathbf{x}^l - \mathbf{x}^{l-1}\| & \text{otherwise.} \end{cases} \tag{26}$$

Let $l = l_j$, we have $\lim_{j \to \infty} \|\partial \Phi(\mathbf{x}^{l_j})\| = 0$, i.e., $0 \in \partial \Phi(\mathbf{x}^*)$. Thus, $\mathbf{x}^*$ is a critical point of $\Phi$.

Finally, we should to prove that $\{\mathbf{x}^l\}_{l \in \mathbb{N}}$ converges to a critical point of $\Phi$, which means that $\{\mathbf{x}^l\}_{l \in \mathbb{N}}$ is a Cauchy sequence. Since $\Phi$ is a semi-algebraic function, it satisfies the KŁproperty. Thus we have that $\sum_{l=1}^{\infty} \|\mathbf{x}^l - \mathbf{x}^{l-1}\| < \infty$ following [1] and Eq. (23). This implies that $\{\mathbf{x}^l\}_{l \in \mathbb{N}}$ is a Cauchy sequence and globally converges to the critical point of $\Phi$. $\qquad \square$

## Footnotes

\*Corresponding Author. Correspondence to `<rsliu@dlut.edu.cn>`.

[2]Actually, a variety of functions in application areas, including $\ell_0$-norm, rational $\ell_p$-norms (e.g., $0 < p \le 1$ and $p$ is rational) and their finite sums or products are all semi-algebraic functions.