[Reviews · NeurIPS 2018]

Reviewer 1



Paper summary: The paper proposed a learning based hybrid proximal gradient method for composite minimization problems. The iteration is divided into two modules: the learning module does data fidelity minimization with certain network-based priors; consequently the optimization module generates strict convergence propagations by applying proximal gradient feedback on the output of the learning module. The generated iterates were shown to be a Cauchy sequence converging to the critical points of the original objective. The method was applied to image restoration tasks with performance evaluated. Comments: The core idea is to develop a learning based optimization module to incorporate domain knowledge into conventional proximal gradient descent procedure. One issue with such a hybrid scheme is that by definition the output of the learning module is acceptable to the proximal gradient module only when the bounding condition in Eq.(9) is met. This then raises a serious concern that the learning module could have very limited effect on the iteration procedure if its output is frequently rejected by the proximal gradient module, i.e., the bound (9) is frequently violated. If that is the case, then the proposed method is no more than a proximal gradient method. So it will be helpful to provide more insightful explanations on the condition (9), e.g., to discuss how hard is it to be fulfilled. Another main concern is the readability of paper. The model description part in Section 3 is rather abstract and hard to follow in general. To gain better intuition of method, the reviewer suggests providing one illustrating example or two such as those appeared in Section 5 to aid model elaboration. The convergence analysis falls on the weak side: the procedure was only shown to converge asymptotically to critical points, using fairly standard proof skills in proximal gradient analysis. The rate of convergence remains largely unknown, even for convex cases. Strong points: + The attempt to bridge model optimization and deep propagation is somewhat interesting and novel. + Numerical study is extensive. Weak points: - The motivation and significance of this work are not fully justified in text and mathematical analysis. - Some important technical details about the proposed algorithms are missing (see the comments above). - The writing quality needs to be significantly improved. === Updated review after author response === I further checked the technical proofs in more details and spot a seemingly fatal flaw in the proof of Proposition 1 (Line 46 - 47 of the supplement): i$x_G^l$ is claimed as a global minimizer of (10). However, as far as I can tell, $x_G^l$ is no more than a first-order stable point of (10). Since the primal interest of this work is on nonconvex problems, it is obvious false to claim that a stable point is a global minimizer. In the meanwhile, after reading the author response, I found most of my major concerns were not clarified in a satisfying way. Particularly, the following issues are still there: 1. The problem statement is quite confusing: the original optimization problem is defined in Eq. (1) but the algorithms are designed for solving (2). The connection between these two formulations is not well explained. In particular, the inner product error term in Eq. (2) does not make sense to me. 2. Unless the authors can show that the bounding condition (9) is eventually always true in iteration, the proposed algorithm essentially reduces to a standard PG descent procedure. 3. The convergence analysis turns out to be weak and of limited novelty: it mimics largely the conventional PG analysis without introducing any conceptually new proof techniques and/or results. All in all, I still have serious concerns about this submission and vote strongly for reject.

Reviewer 2



This paper proposed a Propagative Convergent Network (PCN) to bridge the gaps between model optimization and deep propagation. Different from most existing unrolling approaches, which lack theoretical guarantees, the convergence of PCN is theoretically proved. These convergence results are solid and convincing for me. A relaxed PCN is also proposed to perform end-to-end collaborative learning.   It seems that this work actually addressed an important issue in learning and vision areas. That is, it provided a theoretically guaranteed manner to integrate the advantage of optimization models and heuristic networks for challenging learning tasks.   In the experimental part, authors compared PCN and relaxed PCN with state-of-the-art algorithms on different applications. These results seem convincing for a theoretical paper.   Some minor problems: 1.      In relaxed PCN, the network architecture is not clear. 2.      The training details should be explained in the experimental part. 3.      The parameter setting in experiments should also be given. Such as “p” in sparse coding and “a, b” in image restoration. ######################After Rebuttal ################################ Reviewer 2 spot a mistake in Eq.(10) .I also noticed the minor mistake in Eq.(10) on supplemental materials. In non-convex problem, it should be $\in$ not “=”. The author seems to build a minimize problem as Eq.(10) so that $x_g^l$ is one of the minimizers by introducing $\epsilon$. I think it is a constructing proof strategy. Obviously, this constructed optimization can make the inequality in (11) holds since $x_g^l$ is one of the minimizers. I also noticed the similar mistake happened in Eq.(12). It also should be $\in$ rather than $=$. But it doesn’t affect the deduction from Eq.(12) to (13). Similar results could be found from the traditional nonconvex optimization paper e.g.“Proximal Alternating Linearized Minimization for Non-convex and Non-smooth Problems ”. I think these minor mistakes don’t affect the final convergence conclusion. About the effectiveness of bounding condition, Fig.2(d) in manuscript partially shown the condition is always satisfied in their experiments. I do agree that, from the optimization perspective, this paper suffers from somewhat limited novelty. However, from the deep learning perspective, this paper integrates deep learning module with the non-convex proximal gradient methods, which bridges the network and traditional iteration. Thus it makes sense for deep learning methods. I think it provides a significant evidence for the issue that deep learning strategies can learn a better descent gradient in a meta-learning-like manner. From the computer vision perspective, we see the proposed method could be successfully applied to low-level computer vision tasks such as super-resolution based sparse coding and image restoration. Overall, from deep learning and computer vision perspective, I think this paper is qualified, I will insist on my initial choice.

Reviewer 3



This paper proposed a propagative convergent network that bridges between model optimization and deep propagation in a collaborative manner, where the PCN performs as a deep trainable solver used for optimization. The key idea is to use a generic model-inspired architecture and a proximal feedback mechanism to check and update until converging. The paper proves PCN to have strict convergence property, where a relaxed one can provide an end-to-end learning framework. This paper is well written and easy to understand. The theories are clear and adequately presented in both the paper and the supplementary material (I roughly checked through every step). One concern I have is the experimental part for image restoration. Although it is not specifically targeting on vision tasks, many pieces of information are not clearly presented, including the task settings (e.g., is it a super-resolution with known degradations, or blind/unblind deblur?), some brief introductions of related work, and most importantly, the insights for why the proposed model can achieve better results, e.g., with either theoretical explanation, or experimental ablation studies. Given the above paper strength and concerns, I would give the paper in the current version as score 8. I am not an expert in this area, but I can roughly understand all the deductions and proves. I really prefer the collaborative mechanism shown in (9) and (10), as well as the convergence property of the framework. I hope the authors can address the concern regarding to the experiments. I would also see how other reviewers would comment given more expertise knowledge. Update: I would retain my previous rating after reading the rebuttal, I agree that the issue raise by other reviewers with eq 10 is not a major claim and thus not a factor to re-rate.